# Cognition and Consciousness Entwined

**DOI:** 10.3390/brainsci13060872

**Published:** 2023-05-28

**Authors:** Peter Grindrod, Martin Brennan

**Affiliations:** Mathematical Institute, University of Oxford, Oxford OX2 6GG, UK

**Keywords:** phenomenological sensations, architecture, delay dynamics, modes, preconditioning, simulations, AI, non-binary chips

## Abstract

We argue that cognition (information processing) and internal phenomenological sensations, including emotions, are intimately related and are not separable. We aver that phenomenological sensations are dynamical “modes” of firing behaviour that (i) exist over time and over large parts of the cortex’s neuron-to-neuron network and (ii) are consequences of the network-of-networks architecture, coupling the individual neuronal dynamics and the necessary time delay incurred by neuron-to-neuron transmission: if you possess those system properties, then you will have the dynamical modes and, thus, the phenomenological sensations. These modes are consequences of incoming external stimuli and are competitive within the system, suppressing and locking-out one another. On the other hand, the presence of any such mode acts as a preconditioner for the immediate (dynamic) cognitive processing of information. Thus, internal phenomenological sensations, including emotions, reduce the immediate decision set (of feasible interpretations) and hence the cognitive load. For organisms with such a mental inner life, there would clearly be a large cognitive evolutionary advantage, resulting in the well-known “thinking fast, thinking slow” phenomena. We call this the *entwinement hypothesis*: how latent conscious phenomena arise from the dynamics of the cognitive processing load, and how these precondition the cognitive tasks immediately following. We discuss how internal dynamical modes, which are candidates for emotions down to single qualia, can be observed by reverse engineering large sets of simulations of system’s stimulated responses, either using vast supercomputers (with full 10B neuronal network analyses) or else using laptops to do the same for appropriately generalised Kuramoto models (networks of *k*-dimensional clocks, each representing the 10,000 neurons within a single neural column). We explain why such simplifications are appropriate. We also discuss the consequent cognitive advantages for information-processing systems exhibiting internal sensations and the exciting implications for next-generation (non-binary) computation and for AI.

## 1. Introduction

In direct response to external events, which stimulate the brain with an incoming collage of stimuli via our sensory organs, or to our present train of thought, or to both at the same time, we experience a wide range of internal phenomenological sensations repeatedly and consistently.

In a complementary narrative to that set out in [1], we argue that these inner sensations have a hugely practicable role: they are not an independent or a marginal “extra”. Instead they precondition our immediate subconscious and conscious thinking; they reduce (constrain) the range of the immediate possibilities that need to be considered or managed without us knowing; thus they allow our minds to zoom-in and focus on the most essential matters at hand. We will discuss how, under those stimuli, the brain responds with one of a number of internal *dynamical modes*, which, in turn, affect its information processing operations (cognition) and which reduce the cognitive load for our decision-making. Thus, each of us makes just the sort of decisions that we always make in various circumstances consistently though not necessarily rationally [2]. We do so regardless of any predefined or objective optimality: that is the just sort of person that we are. It would take some hard, conscious effort to recognise ourselves doing so and to pause, back-up, and break our usual chain of response and consequent action [3].

Inner phenomenological sensations obviously differ in their scale: they can be large or small. They can be almost overwhelming emotions such as love, lust, fear, grief, anxiety, ecstasy, pain, or embarrassment; or they can be small experiences, such as the feelings of seeing the blueness of blue, or of the stroking of skin, or hearing the sound of a trumpet. The latter are often called “qualia” and are accessible phenomenal components of our mental lives. As discussed in [4], they can be brought to mind by thinking of a collage of instances, past events, images, or music: you can make yourself feel happy, sad, or even embarrassed. Hence, they cannot be reliant on any dynamical instability or emergence phenomena.

We might also posit that these inner sensations are hierarchical, with some being components of others. That is not a new idea. Yet, as we shall see, that accords very well with our attempts to reverse engineer simulations of the human brain and observe its dynamical response to stimuli (see Section 3.2). By “dynamical response”, we mean physical modes of neural activity that exist over both time and across the brain. They are not stationary and they cannot be classified in a single snapshot. In Section 3.2, we will discuss how we may observe these common response modes of dynamical behaviour within a cortex-like cortex system via multiple experiments where the system is subject to all kinds of stimuli. The discovery of these distinct internal dynamical modes, which are prime candidates for internal sensations, requires a kind of in silico reverse engineering.

The simulations considered in Section 3 are designed with the brain’s specific network architecture of neural connections in mind: one that is very far from random. Indeed, we shall see why this is efficient in an evolutionary sense. The architecture is introduced briefly in Section 2. Network science is an extremely fast moving area of research, yet the very nature of cognition and consciousness within the brain provide hard challenges to explain how such models might reflect early development (the dynamical formation of networks), plasticity (how cognitive usage reinforces the network and yet is constrained by the network), and possible phase changes when network properties pass hidden thresholds, possibly resulting in a huge increase in the emotional (sensational) inner life as well as a plethora of expanded types of thinking.

There is an insightful discussion of some (dynamical) network properties of the brain presented in [1]. There it is argued, much as here, that emotional processing appears to be interlocked with cognitive processing, and network science is the key (to both the dynamics of the developing networks and the activity dynamics supported by the network). More generally, [1] considers how we should think of causation within complex systems in order to inform that relationship between emotion and cognition in the brain.

## 2. The Architecture of the Cortex and Associated Delay-Dynamical Systems

The human cortex contains around 10B neurons. They are arranged into approximately O(1M) neural columns, with each neural columns containing O(10,000) rather densely connected neurons. The columns are arranged in a two-dimensional grid over the surface of the cortex. If the human cortex was stretched out flat, un-corrugated, it would be like a carpet with the separate columns forming the carpet pile. Some neurons within near-neighbouring columns are connected (all connections are directional). All neuron-to-neuron transmission of firing spikes (signals) incurs an individual (real valued) time delay (the time taken for a signal spike of membrane potential to leave the sender’s soma, travel out along its axon, cross a synapse, and then travel inwards via a dendrite of the receiving neuron, reaching its soma). All the connections are directional. The outer column-to-column networks is *range dependent* in that columns located at a further and further distance (range) are less likely to have any direct connections (depicted flattened out in Figure 1).

Hence, the cortex has a *network-of-networks* architecture, with the inner networks representing the densely connected neurons within neural columns, and the *range-dependent* outer network made form connections between neurons from near-neighbouring columns. The architecture is really very important for two reasons. First, we can examine how the dynamics of an individual column behave by direct simulations of its densely coupled neurons [5]. Second, we may exploit the *network-of-networks* architecture in both (a) full very-large-scale (VLS) simulations of all neuron-to-neuron connections [6] modelling an excitatory–refractory dynamic at every neuron and incorporating random transmission time delays (requiring supercomputers) and (b) motivating dynamical simplifications of the whole to generalised Kuramoto networks, where each neural column is represented by a summary dynamical system equivalent to a high-dimensional generalised clock (see Section 3.1) coupled within the outer network, whence the simulation of the whole is thus tractable on a normal laptop [7].

## 3. What Do We Learn from Very-Large-Scale Simulations?

### 3.1. Individual Neural Column Simulations

The VLS neuron-to-neuron simulations in [5] for individual neural columns revealed that each column actually behaves like a *k*-dimensional clock, and that *k* is observed to be proportional to the log of the number of neurons within a single column [5]. A *k*-dimensional clock has *k* independent phases, each of which winds around (mod 2π) at an individual rate. In analysing simulations, the number of degrees of freedom, *k*, exhibited in a neuron-to-neuron simulation of the individual neural column may be directly estimated via state-space embedding techniques applied to neuronal spike trains (see [5] and the references therein). Note that *k* does not simply count the number of cycles within a directed network since not all of those are viable (given the refractory nature of neuron spiking) and some cycles would cancel out one another. It might be better to think that *k* counts the longish, yet mutually independent, cycles. The oscillations arise from these cycles: no individual neurons are in an oscillatory regime (instead, they are all both excitable and refractory).

Importantly, [5] shows that *k* grows similar to logn, where *n* is the number of neurons within a column. This simple fact explains why the brain has evolved to have many columns across the cortex of a rather uniform size rather than a wide distribution of columns of different sizes (different *n*): the latter would waste both energy and volume (two crucial constraints on the human brain) since, in terms of total degrees of freedom of the dynamics, it would be better to have two distinct columns (of a minimum viable size) rather than just one of double the size.

This fact underpins the evolutionary advantage of the uniform network-of-networks architecture that we observe. Typically, it is estimated that k∼O(10) for n=10,000 or so neurons within a single column.

### 3.2. Whole Cortex Simulations

In [6], full VLS neuron-to-neuron model simulations of the whole cortex were made utilising a supercomputer (the 1M-core Spinnaker). Once specified, the whole system was subject to incoming stimuli (forcing terms) in the form of trains of incoming spikes applied at various specific neurons. Many independent experiments were carried out within different forcings. Unlike a real brain, it is possible to *reverse engineer* the system to see what occurred within under various different forcing regimes. This was achieved by having a few “observed” neurons within every column for which the firing sequence was recorded. Consequently, within each experiment, a dynamical response in time and across all of the neural columns was observed. Many such experiments were carried out, applying the forcing at different neurons selected from across all of the columns. These calculations were a massive undertaking: in essence, analysing the whole-system response represents a big-data problem itself. What was discovered?

It turned out [6] that by considering more than 1000 separate experiments, the corresponding internal dynamical responses (defined over time and across the large observed set of neurons) could be clustered. This was shown to be significant (since whenever you look for clustering, you will find some even where there should be none). The pairwise comparison between more than 1000 responses needs to allow some absolute time offsets, since it is the relative firing patterns that are important, not absolute time. The hierarchical clusters represent distinct classes of similar dynamical internal response (of the system to the forcing regime). Each cluster corresponds to a dynamic response “mode”. Any similar stimulus will likely result in one or another mode kicking in. These dynamical modes are good candidates for internal phenomenological sensations arranged hierarchically for large-scale emotions down to small-scale qualia. Each sensation corresponds to the internal mode becoming active. Furthermore, at any level of the hierarchy, the modes are competitive and do not co-exist.

All of this accords with the view set out in [1] based on findings from pattern analyses of neuroimaging data that show that affective dimensions and emotion categories can be detected in the activity of distributed neural systems that span cortical and subcortical regions.

So right there inside the dynamical system is a possible material basis for sensations and emotions. While they are consequences of the incoming forcing (external experience), they can also precondition the brain’s response to the immediately incoming signals. While the cognitive challenge is to answer the question “What is happening now?”, the conscious response (the dynamical modes) can precondition the brain in answering that question. Thus, *love may be blind* (at least it is blind to some possibilities). The restriction of the feasible decision space through such preconditioning would confer massive efficiency advantages over a non-preconditioned brain, where anything might be possible at any time. Thus, there would be an evolutionary advantage to the two-way coupling between cognition and consciousness. We will return to this point in Section 4.

In [7], the authors consider a different type of whole-cortex model and simulations, one that is much simpler and which is feasible to run on a laptop. It is based on the network-of-networks architecture and the insights concerning neural columns, as follows. Each neural column is represented by a *k*-dimensional clock. These are coupled by directed connections within a range-dependent network. For each directed connection, whenever the phases of a sending clock (neural column) reach a certain (edge-dependent) trigger condition, it sends out a signal to the receiving clock (neural column). The signal incurs an edge-dependent time delay, and when it arrives at the receiving clock, it applies an instantaneous (edge-dependent) phase-resetting map (PRM) to the phases of the receiving clock. Coupled systems with an array of simple clocks are usually called a *Kuramoto system*: this particular system represents a generalisation to consider coupled *k*-dimensional clocks set within a range-dependent network and using instantaneous PRMs as the clock-to-clock coupling mechanism. The generalised Kuramoto system contains about 10M state variables (say 10 phases for each of the 1M clocks/neural columns), whereas a full neuron-to neuron simulation would have at least 10B state variables (one phase for each neuron). Hence, it can be run on a much more modest computing platform.

Even making pragmatic simplifying assumptions, simulations employing this type of system may be tested in the same way as the full-neuron simulations in [6]. In [7], it is shown that the whole system of coupled clocks exhibits internal, hierarchically arranged, responsive dynamical modes, just as the full simulations did. By carrying out a rather large number of similar experiments, one can reverse engineer the generalised Kuramoto system, just as was done for the full simulations in [6]. Again, inside the Kuramoto dynamical system, we observe a possible material basis for sensations and emotions.

Of course, both of these types of dynamical simulation exhibit dynamical modes that are very good candidates for phenomenological sensations. The sensation is the brain’s own experiences of the corresponding mode being present (active) in the moment. There is still an explanatory gap though: we cannot be sure that the dynamical modes cause those sensations, yet they have not been observed before and hence they are not discussed within the philosophical or brain-science literature. They are behavioural modes distributed across the cortex and across time.

Indeed, early philosophical work often argued that cognition and consciousness are separate, or that cognition begets consciousness as a consequence or by-product (see [8], for example). However, here we suggest that we should accept the corollary (to the insights from the simulations) that internal conscious phenomena are crucial to certain efficiencies within cognition. Cognition and consciousness would be, thus, mutually dependent and entwined.

The situation is summarised in Figure 1.

## 4. Implications for Next Generation AI

Present AI and, in particular, deep learning, has reached an inflection point. There are now many successful applications of automated classifiers (addressing supervised recognition and discrimination problems) driven by powerful modern computation, accomplishing otherwise tedious, repetitive, parallel discrimination tasks at scale. These algorithms *emulate* human perception and discrimination performance (they certainly do not work in the same way as a human brain), yet they can be deployed with a bandwidth that is far beyond any human capabilities.

So, what can we learn from the concepts and mechanisms that appear to be present within consciousness that might extend or improve the existing approaches to AI?

Of course, an AI application does not posses a reservoir of common sense: it has only seen what it has been shown during calibration, and it has no abstracted or generalised common knowledge. Thus, the AI algorithm does not rule out irrelevant information within the input. For example, within real-world image analysis, there may be events or objects in the far background or the near foreground that are irrelevant to the task at hand, but we simply do not know what the algorithm will focus on, and, indeed, it has no concepts of background and foreground and so on. So perhaps the analogy of internal modes (the equivalent of sensations or qualia) as a preconditioner for cognitive tasks would be very useful (if it ever were implementable). The idea is that outputs should be conditional not just on the externally presented inputs but also upon the internal (latent) state of the system. Although making the AI less consistent, with outputs depending on the (current, real time) state (of its internal sensory experience), the state may bias and heavily constrain outputs (reducing the decision set), so that the whole becomes more blind to irrelevancies. This would lead to much more permissive classification and inference algorithms that need to be in the right *mood* in certain circumstances (an issue for operators). Nevertheless, performance might actually become more robust and unspoofable (due to small irrelevancies in the inputs) though, admittedly, still lacking of any reservoir of common sense.

The concept of internal sensations acting as preconditioners for cognitive processing and response (represented temporal latent modes or other phenomena) may become crucial in enabling new models of AI. This appears similar in spirit to underpinning transformer models, which introduced a human cognitive concept: *attention* [9]. Consciousness is, arguably, more fundamental and more central to human information processing than attention, so much may be gained by exploring AI models that encompass these ideas.

## 5. Implications for Neuromorphic Non-Binary Chips

Ideas to develop brain-inspired computer chips are not new. In industry, IBM’s TrueNorth chip [10,11], for example, represented a major advance as part of a long-term DARPA program to develop neuromorphic machine technology and build a new kind of cognitive computer. This led to a patent application [12] that revealed some of the inner workings of TrueNorth. Other labs have also designed neuromorphic chips—such as Intel’s Loihi chip, which is positioned as emulating human brain function and probabilistic computing [13]—aiming to instantiate a spiking neural network at the individual neuron level and upwards (130,000 neuron-equivalents to 13 neural columns). There are also programmes at various universities and national labs, usefully summarised recently in [14].

These developments have been ground-breaking in parts, and their impact on industry has made a huge difference already. However, it is important to note that they also face some limitations of both a practical and theoretical nature.

One possible limitation of IBM’s TrueNorth, for example, may be observed via direct simulations: it updates on the tick of the clock. If we model a (small) cortical column, then TrueNorth is equivalent to assuming that all neuron-to-neuron connections have time delays associated with the same value (unity). We carry out that computing experiment, with any reasonable (excitatory and refractory) spiking dynamics, by first deploying real-valued time delays, whereby we observe that the whole behaves as a winding map with *k* degrees of freedom (*k* phases), just as in [5] and discussed above. When we force all of the time delays to equal unity, then the achieved value for *k* quenches, becoming massively reduced. This is because alternative directed walks from neuron A to neuron B of the same length must arrive at the same time (there are many dead heats). This fact appears never to have been discussed and is a foundational stumbling block: the discreet uniform time-step updates are fundamental to the chip.

The obvious challenge is to design a mathematical model of the human brain’s own information-processing mechanism that utilizes not only logic but, specifically, the power and effectiveness endowed by dynamical, latent, phenomenological preconditioning.

For example, considering the earlier challenge for VLS high-resolution simulation. One ansatz for this challenge would be to fabricate an array of *N* individual *k*-dimensional clocks (called *k*-clocks, with k>1), with directed connections between near-neighbouring clocks, embedded within a two-dimensional grid (as in [7]).

This is just one of a number of alternative paradigms that are suggested by whole-brain simulation and its abstraction in the context of consciousness. In all cases, the challenge would not only be one of mathematical design but also one of instantiating them in an appropriate physical way: not as code running on binary processing platforms, which VLS simulations have already shown to be prohibitively expensive and very slow (even on massive multi-core platforms). Hence, the urgent need for neuromorphic chips. On the mathematical side of things, this implies not resolving the typical equations used within simulations; instead, we must aim to use models of appropriate physical (perhaps organic) materials and architectures that are naturally well-suited to the tasks (just as the neural columns and the whole cortex are, having been shaped to be so by evolutionary forces). Additionally, we should deploy mathematics to *verify* that we have achieved this.

We should condition (train) these whole system’s details so as to produce distinguished output inferences and decisions in response to some distinct classes of inputs.

Note that, unlike binary logic-based information processing, it is very likely that such a system may (similar to the human brain) be rather poor at logical and arithmetic tasks but good at inferring fast decisions from incomplete data. We should not ask it to perform intensive signal-processing tasks. Instead, we should challenge it with multi-way discrimination and decision-making tasks and reward novelty (and discovery) rather than optimising any objective functions.

## 6. Discussion

We have highlighted a number of distinct challenges and consequences. First, there is no real need to make full neuron-to-neuron simulations. What we learn from these is that individual neural columns (in isolation) behave like *k*-dimensional clocks, which are just winding maps over a *k*-dimensional torus. The *k* phase coordinates of the torus are equivalent to the phases of separate independent cycles embedded within the directed network connections of the columns. This result alone anticipates the relatively uniform nature of the cortex: brains with a large distribution of column sizes make inefficient use of both energy and volume in maximising the total number of degrees of freedom across all of the columns. This observation also suggests that simulations may be carried out by exploiting the network-of-networks architecture with the generalised *Kuramoto system* with an array of coupled *k*-dimensional clocks set within a range-dependent network and using instantaneous PRMs as the clock-to-clock coupling mechanism (as in [7]).

Moreover, when we reverse engineer these simulations, we find a hierarchy of dynamical behavioural modes that are candidates for the physical basis of internal sensations: emotions down to qualia. As a result, when such modes are present, any immediate incoming cognitive task (making use of sensory inputs) is constrained to have a much smaller decision space, and this endows the brain with a fast-thinking advantage. Thus, emotions (conscious phenomena) are entwined with information processing (cognition). You cannot have one without the other.

This fact suggests many questions about the brain’s development from infants to adolescents. As the connection density increases as a result of experience, the brain not only develops more-subtle or intricate cognitive abilities, but it necessarily develops a more-sophisticated palate of internal phenomenological sensations. Without the latter, the former would cause the brain to be ponderous and ineffective (and only slowly responsive). Thus, the development of emotional behaviours is necessary for the development of cognitive abilities and vice versa.

Of course, the narrative set out here poses many new problems for mathematical analysts in both modelling and signal processing. It also poses some potential solutions to our understanding of how the brain *may* develop and decline, with cognition and consciousness entwined. As a further result, the processes of coupled cognitive and consciousness instantiation may not only benefit our understanding of the human brain, but, as we have discussed above, it may contribute to much better exploration of novel types of neuromorphic information processing (and in criticising existing ones) as well as novel concepts in next-generation AI.

## Figures and Tables

**Figure 1 brainsci-13-00872-f001:**
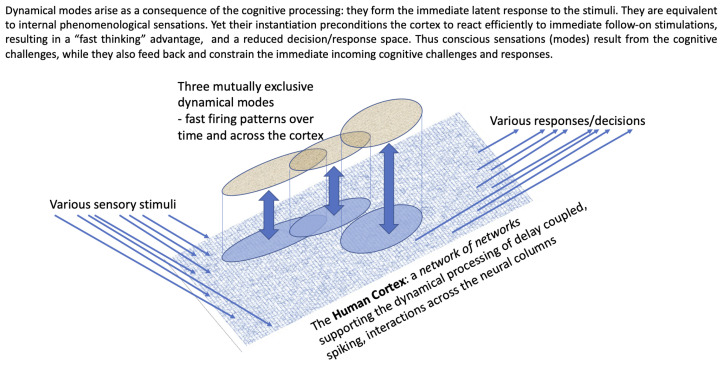
The interaction between conscious sensations (extant dynamical modes) and immediate cognitive processing across the cortex. The stimuli and responses are multiple parallel spike trains. The modes are common patterns of internal dynamical behaviour.

## Data Availability

No data was collected or created during this research.

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
