# Peer review of "Cognition and Consciousness Entwined"

_brainsci, 2023, doi:10.3390/brainsci13060872_

Round 1

Reviewer 1 Report

This is a review article that uses the conclusions of several previous studies to discuss cognition and consciousness. Although the author explains the relevant conclusive concepts by citing and elaborating on several previous studies, the citations are simple and the exposition is relative general. 

In addition, the methods and data used in previous studies to reach these relevant conclusions still need to be improved. As such important points need to be elaborated and reflected, the missing information may affect the understanding and interest of reader in the article. 

Moreover, if the relevant information or the stage process of the research (including the results of references and previous studies) can be shown using charts, tables, and figures, it will be more attractive to readers, which is also missing in this paper.

Author Response

REVIEWER 1

This is a review article (It is not a review) that uses the conclusions of several previous studies to discuss cognition and consciousness. Although the author explains the relevant conclusive concepts by citing and elaborating on several previous studies, the citations are simple and the exposition is relative general. 

This article was rather hard to write for three main reasons. The reviewer needs to consider it in the light of the following points.

  1. It is not a general review: it summarises very recent discoveries from computational simulations for the whole cortex using mathematical models (based on the network-of-networks architecture, the directed neuron-to-neuron time-delayed spike transmissions, and the excitable/refractory nature if single neuron dynamics). In doing so we have sought to avoid the use of mathematical nomenclature or equations and instead to describe the activities and discoveries, and yet explain how it is that conscious phenomena (internal emotions and feelings) may be contributing to cognitive performance, and vice versa (a two-way fast-thinking evolutionary advantage.
  2. We recognised that the invitation to publish such a narrative in this special issue would need us to address a multidisciplinary audience, and that presenting/discussing the summarised results for very large-scale simulations would be a challenge. As a result the article sets out a clear hypothesis (the “entwinement” hypothesis), and explains the objective facts on which it is based; it explains the possible evolutionary advantage arising; and it moves the reader forwards to consider the implications for next-generation computation (non-binary chips) and next-generation AI. It is thus an exposition of the “what” and “why” for future research and invites discussion and possible push-back from a wide variety of disciplines. It will stimulate debate.
  3. We normally write for a mathematical and computation audiences: we assumed that the invitation to contribute here, to the special issue, was designed to allow multidisciplinary interaction and for the work on simulation to inform other disciplines working in the space. We have introduced appropriate concepts – some, such as multi-dimensional clocks (dynamics on high dimensional tori) or methods of reverse engineering a large ensemble of simulations will be unfamiliar to most of the journal readers. We thus needed to be clear and necessarily selective. It is thus a clear statement of this “entwinement” hypothesis that is important.

In addition, the methods and data used in previous studies to reach these relevant conclusions still need to be improved. As such important points need to be elaborated and reflected, the missing information may affect the understanding and interest of reader in the article. 

The article set out the “entwinement” hypothesis as an assessable narrative.

Moreover, if the relevant information or the stage process of the research (including the results of references and previous studies) can be shown using charts, tables, and figures, it will be more attractive to readers, which is also missing in this paper.

We will not illustrate the text with unnecessary figure or charts.  Readers van pick up the narrative, the “entwinement” hypothesis,  and can consider if it is objectionable or attractive to them.

Have made appropraite corrections

Reviewer 2 Report

The present research article by Grindod entitled ‘COGNITION AND CONSCIOUSNESS ENTWINED’ is a well-written and useful summary on the current status of knowledge of the use of engineering large sets of simulations to properly simulate information processing systems exhibiting internal sensations.

In general, I think the idea of this article is really interesting and the authors’ fascinating observations on this timely topic may be of interest to the readers of Brain Sciences. However, some comments, as well as some crucial evidence that should be included to support the author’s argumentation, needed to be addressed to improve the quality of the manuscript, its adequacy, and its readability prior to the publication in the present form, in particular reshaping parts of the Introduction and Methods sections by adding more evidence and theoretical constructs.

Please consider the following comments:

Abstract: According to the Journal’s guidelines, the abstract should be a total of about 200 words maximum and should be presented as a single paragraph, without sub-headings. Please correct the actual one. Also, in my opinion, Authors should consider rephrasing this section. According to the Journal’s guidelines, the Abstract should contain most of the following kinds of information in brief form. Please, consider giving a more synthetic overview of the paper's key points: I would suggest rephrasing the results and conclusion to make them clear for readers to understand.

A graphical abstract that will visually summarize the main findings of the manuscript is highly recommended.

Keywords: I would suggest changing the keyword ‘AI’ into ‘Artificial Intelligence (AI)’, which in my opinion is clearer and more appropriate.

In general, I recommend authors to use more references to back their claims, especially in the Introduction of this research article, which I believe is lacking. Thus, I recommend the authors to attempt to expand the topic of their article, as the bibliography is too concise. Therefore, I suggest the authors to focus their efforts on researching relevant literature: in my opinion, adding more citations will help to provide better and more accurate background to this study. 

Introduction: I would suggest the authors to reorganize the Introduction section, which seems not enough extensive and it does not seem to consider, in most cases, all the available studies in the literature that have acknowledged the complexity of neural correlates of consciousness. In this regard, I think that it would be useful to have more information about how the neural correlates of consciousness should be divided into primary and secondary brain areas, whose early activity in the occipital lobe may support the first perceptual discriminations among stimuli (https://doi.org/10.3390/biomedicines10081897) and later activity in frontal areas like medial prefrontal cortex, which may support the integration of different visual features that are contingent on the outcomes of the earlier perceptual processing (https://doi.org/10.3390/biomedicines10123189).

Discussion: In this final section, authors described the results of their study and their argumentation and captured the state of the art well; however, I would have liked to see some views on a way forward. I believe that the authors should make an effort, trying to explain the theoretical implication as well as the translational application of this paper, to adequately convey what they believe is the take-home message of their study. In this regard, I believe that it would be necessary to discuss theoretical and methodological avenues in need of refinement.

I think that a proper and defined ‘Conclusions’ and ‘Limitations and future directions’ paragraphs would be very useful to convey some thoughtful as well as in-depth considerations by the author. Author should make an effort, trying to explain the theoretical implication as well as the translational application of the research.

References: Authors should consider revising the bibliography, as there are several incorrect citations. Indeed, according to the Journal’s guidelines, they should provide the abbreviated journal name in italics, the year of publication in bold, the volume number in italics for all the references. Also, please correct in-text citations: reference should be numbered, and placed in square brackets [ ] (for example [1]).

I hope that, after these careful revisions, this paper can meet the Journal’s high standards for publication. 

I am available for a new round of revision of this paper. I declare no conflict of interest regarding this manuscript. 

Best regards,

Reviewer

Author Response

REVIEWER 2

The present research article by Grindrod entitled ‘COGNITION AND CONSCIOUSNESS ENTWINED’ is a well-written and useful summary on the current status of knowledge of the use of engineering large sets of simulations to properly simulate information processing systems exhibiting internal sensations.

In general, I think the idea of this article is really interesting and the authors’ fascinating observations on this timely topic may be of interest to the readers of Brain Sciences. However, some comments, as well as some crucial evidence that should be included to support the author’s argumentation, needed to be addressed to improve the quality of the manuscript, its adequacy, and its readability prior to the publication in the present form, in particular reshaping parts of the Introduction and Methods sections by adding more evidence and theoretical constructs.

This article was rather hard to write for three main reasons. The reviewer needss to consider it in the light of the following points.

  1. It is not a general review: it summarises very recent discoveries from computational simulations for the whole cortex using mathematical models (based on the network-of-networks architecture, the directed neuron-to-neuron time-delayed spike transmissions, and the excitable/refractory nature if single neuron dynamics). In doing so we have sought to avoid the use of mathematical nomenclature or equations and instead to describe the activities and discoveries, and yet explain how it is that conscious phenomena (internal emotions and feelings) may be contributing to cognitive performance, and vice versa (a two-way fast-thinking evolutionary advantage.
  2. We recognised that the invitation to publish such a narrative in this special issue would need us to address a multidisciplinary audience, and that presenting/discussing the summarised results for very large-scale simulations would be a challenge. As a result the article sets out a clear hypothesis (the “entwinement” hypothesis), and explains the objective facts on which it is based; it explains the possible evolutionary advantage arising; and it moves the reader forwards to consider the implications for next-generation computation (non-binary chips) and next-generation AI. It is thus an exposition of the “what” and “why” for future research and invites discussion and possible push-back from a wide variety of disciplines. It will stimulate debate.
  3. We normally write for a mathematical and computation audiences: we assumed that the invitation to contribute here, to the special issue, was designed to allow multidisciplinary interaction and for the work on simulation to inform other disciplines working in the space. We have introduced appropriate concepts – some, such as multi-dimensional clocks (dynamics on high dimensional tori) or methods of reverse engineering a large ensemble of simulations will be unfamiliar to most of the journal readers. We thus needed to be clear and necessarily selective. It is thus a clear statement of this “entwinement” hypothesis that is important.Have made appropraite corrections

Please consider the following comments:

• Abstract: According to the Journal’s guidelines, the abstract should be a total of about 200 words maximum and should be presented as a single paragraph, without sub-headings. Please correct the actual one. Also, in my opinion, Authors should consider rephrasing this section. According to the Journal’s guidelines, the Abstract should contain most of the following kinds of information in brief form. Please, consider giving a more synthetic overview of the paper's key points: I would suggest rephrasing the results and conclusion to make them clear for readers to understand.

• A graphical abstract that will visually summarize the main findings of the manuscript is highly recommended.

We will not illustrate the text with unnecessary figure or charts.  Readers can pick up the narrative, the “entwinement” hypothesis,  and can consider if it is objectionable or attractive to them.

• Keywords: I would suggest changing the keyword ‘AI’ into ‘Artificial Intelligence (AI)’, which in my opinion is clearer and more appropriate.

“AI” is very standard these days – no problem here.

• In general, I recommend authors to use more references to back their claims, especially in the Introduction of this research article, which I believe is lacking. Thus, I recommend the authors to attempt to expand the topic of their article, as the bibliography is too concise. Therefore, I suggest the authors to focus their efforts on researching relevant literature: in my opinion, adding more citations will help to provide better and more accurate background to this study. 

This is certainly not intended to be any kind of an exhaustive review – we could have include 100 references. The central point is to describe the entwinement hypotheses and illustrate (with just enough references) how it is supported by the facts of the reverse engineering of very large ensembles of very large scale simulations.

• Introduction: I would suggest the authors to reorganize the Introduction section, which seems not enough extensive and it does not seem to consider, in most cases, all the available studies in the literature that have acknowledged the complexity of neural correlates of consciousness. In this regard, I think that it would be useful to have more information about how the neural correlates of consciousness should be divided into primary and secondary brain areas, whose early activity in the occipital lobe may support the first perceptual discriminations among stimuli (https://doi.org/10.3390/biomedicines10081897) and later activity in frontal areas like medial prefrontal cortex, which may support the integration of different visual features that are contingent on the outcomes of the earlier perceptual processing (https://doi.org/10.3390/biomedicines10123189).

• Discussion: In this final section, authors described the results of their study and their argumentation and captured the state of the art well; however, I would have liked to see some views on a way forward. I believe that the authors should make an effort, trying to explain the theoretical implication as well as the translational application of this paper, to adequately convey what they believe is the take-home message of their study. In this regard, I believe that it would be necessary to discuss theoretical and methodological avenues in need of refinement.

• I think that a proper and defined ‘Conclusions’ and ‘Limitations and future directions’ paragraphs would be very useful to convey some thoughtful as well as in-depth considerations by the author. Author should make an effort, trying to explain the theoretical implication as well as the translational application of the research.

• References: Authors should consider revising the bibliography, as there are several incorrect citations. Indeed, according to the Journal’s guidelines, they should provide the abbreviated journal name in italics, the year of publication in bold, the volume number in italics for all the references. Also, please correct in-text citations: reference should be numbered, and placed in square brackets [ ] (for example [1]).

I hope that, after these careful revisions, this paper can meet the Journal’s high standards for publication. 

I am available for a new round of revision of this paper. I declare no conflict of interest regarding this manuscript. 

Reviewer 3 Report

Dear authors,

I read your work and found it interesting and debatable which is more suitable as a book chapter because it is like an essay not as a scientific article for journals, such as Brain Sciences which requires rigorous data analysis. Therefore, I suggest publishing it as a book chapter or in arXiv.

• What is the main question addressed by the research? The authors discuss that cognition and internal phenomenological sensations, including emotions, are intimately related and are not separable. They discuss what could be a right solution for future generation AI and what advantages and limitations of current AI are. 
  • Do you consider the topic original or relevant in the field? Does it

address a specific gap in the field? It does, but rather a discussion without any fundamental and rigorous analysis (lack of literature review), at least not from what I see in the manuscript.  

• What does it add to the subject area compared with other published
material? Authors argue that there is no real need to make full neuron-to-neuron simulations. They conclude that the development of emotional behaviours is necessary for the development of cognitive abilities and vice versa and for future generation AI.  

• What specific improvements should the authors consider regarding the
methodology? What further controls should be considered? The manuscript is rather a discussion/review and does not carry rigorous statistical analysis nor is it a systematic review. I suggest enriching the references.  

• Are the conclusions consistent with the evidence and arguments presented
and do they address the main question posed? There is no conclusion section that summarizes the manuscript. I encourage authors to state the objective, their findings and recommendations in the conclusion section. Also, they can improve their discussion section by comparing their findings in light of other similar research.  

• Are the references appropriate? In my view, the references are not enough. The authors talk about emotion classification, deep learning methods, future generation AI, but without adequate references. For example, in line 154. The authors can discuss current ongoing research for emotion classification: https://doi.org/10.3390/s22062346
https://doi.org/10.3390/sym12010021
Therefore, I suggest authors include at least 20 more recent articles to adequately cover different subjects.  

• Please include any additional comments on the tables and figures. There are no figures and tables in the manuscript. I suggest authors include at least one figure like a graphical abstract (for example showing the brain, computers, AI, etc.). A table that summarizes most recent research/articles on human brain and AI methods with very brief discussion on advantages and limitations of each research would enrich this manuscript.   Overall, the manuscript needs significant improvement to meet the requirements of the journal.

Author Response

REVIEWER 3

Dear authors,

I read your work and found it interesting and debatable which is more suitable as a book chapter because it is like an essay not as a scientific article for journals, such as Brain Sciences which requires rigorous data analysis. Therefore, I suggest publishing it as a book chapter or in arXiv.

This article written in response to an invitation to take part in this special issue. It was rather hard to write for three main reasons. The reviewer needs to consider it in the light of the following points.

  1. It is not a general review: it summarises very recent discoveries from computational simulations for the whole cortex using mathematical models (based on the network-of-networks architecture, the directed neuron-to-neuron time-delayed spike transmissions, and the excitable/refractory nature if single neuron dynamics). In doing so we have sought to avoid the use of mathematical nomenclature or equations and instead to describe the activities and discoveries, and yet explain how it is that conscious phenomena (internal emotions and feelings) may be contributing to cognitive performance, and vice versa (a two-way fast-thinking evolutionary advantage.
  2. We recognised that the invitation to publish such a narrative in this special issue would need us to address a multidisciplinary audience, and that presenting/discussing the summarised results for very large-scale simulations would be a challenge. As a result the article sets out a clear hypothesis (the “entwinement” hypothesis), and explains the objective facts on which it is based; it explains the possible evolutionary advantage arising; and it moves the reader forwards to consider the implications for next-generation computation (non-binary chips) and next-generation AI. It is thus an exposition of the “what” and “why” for future research and invites discussion and possible push-back from a wide variety of disciplines. It will stimulate debate.
  3. We normally write for a mathematical and computation audiences: we assumed that the invitation to contribute here, to the special issue, was designed to allow multidisciplinary interaction and for the work on simulation to inform other disciplines working in the space. We have introduced appropriate concepts – some, such as multi-dimensional clocks (dynamics on high dimensional tori) or methods of reverse engineering a large ensemble of simulations will be unfamiliar to most of the journal readers. We thus needed to be clear and necessarily selective. It is thus a clear statement of this “entwinement” hypothesis that is important.

  • What is the main question addressed by the research? The authors discuss that cognition and internal phenomenological sensations, including emotions, are intimately related and are not separable. They discuss what could be a right solution for future generation AI and what advantages and limitations of current AI are. 

  • Do you consider the topic original or relevant in the field? Does it
    address a specific gap in the field? It does, but rather a discussion without any fundamental and rigorous analysis (lack of literature review), at least not from what I see in the manuscript.   

It is not a review: we are setting out an hypothesis.

  • What does it add to the subject area compared with other published
    material? Authors argue that there is no real need to make full neuron-to-neuron simulations. They conclude that the development of emotional behaviours is necessary for the development of cognitive abilities and vice versa and for future generation AI.  
  • What specific improvements should the authors consider regarding the
    methodology? What further controls should be considered? The manuscript is rather a discussion/review and does not carry rigorous statistical analysis nor is it a systematic review. I suggest enriching the references.  

Economy is the name of the game here.

  • Are the conclusions consistent with the evidence and arguments presented
    and do they address the main question posed? There is no conclusion section that summarizes the manuscript. I encourage authors to state the objective, their findings and recommendations in the conclusion section. Also, they can improve their discussion section by comparing their findings in light of other similar research.  
  • Are the references appropriate? In my view, the references are not enough. The authors talk about emotion classification, deep learning methods, future generation AI, but without adequate references. For example, in line 154. The authors can discuss current ongoing research for emotion classification: https://doi.org/10.3390/s22062346
    https://doi.org/10.3390/sym12010021
    Therefore, I suggest authors include at least 20 more recent articles to adequately cover different subjects.   
  • Please include any additional comments on the tables and figures. There are no figures and tables in the manuscript. I suggest authors include at least one figure like a graphical abstract (for example showing the brain, computers, AI, etc.). A table that summarizes most recent research/articles on human brain and AI methods with very brief discussion on advantages and limitations of each research would enrich this manuscript.   Overall, the manuscript needs significant improvement to meet the requirements of the journal.

Round 2

Reviewer 1 Report

Through the author's reply, some of my previous doubts have been dispelled. Please check English writing and grammatical structure to make it more easier to understand.

Author Response

.

Reviewer 3 Report

Dear authors,

I appreciate your explanations, but I could not see any attempt for revising the manuscript.

Regards,

Author Response

.